# MRExtrap: Linear Prediction of Brain Aging in Autoencoder Latent Space of MRI Scans

**Jaivardhan Kapoor**[1]                                   JAIVARDHAN.KAPOOR@UNI-TUEBINGEN.DE
**Jakob H. Macke**[1,2]                                      JAKOB.MACKE@UNI-TUEBINGEN.DE
**Christian F. Baumgartner**[1,3]                      CHRISTIAN.BAUMGARTNER@UNILU.CH

[1] *Machine Learning in Science, University of Tübingen & Tübingen AI Center, Tübingen, Germany*
[2] *Department Empirical Inference, Max Planck Institute for Intelligent Systems, Tübingen, Germany*
[3] *Faculty of Health Sciences and Medicine, University of Lucerne, Switzerland*

## Abstract

Longitudinal generative modeling of high-resolution 3D Magnetic-Resonance-Imaging (MRI) scans can reveal disease progression patterns in neurological disorders such as Alzheimer's disease. We introduce a novel approach called MRExtrap for simulating aging in brain MRI volumes given previously observed MRIs, by performing linear regression in the latent space of an autoencoder. We show that well-trained convolutional autoencoders can yield latent representations that exhibit linearity with respect to the regional brain volumes when interpolated, decoded, and segmented. We exploit this structure by training a linear progression model in the latent space of the autoencoder to predict trajectories of latent representations based on the age of the subject. On the ADNI dataset, we show that predicted MRIs align closely with held-out longitudinal scans, enabling accurate modeling of age-related structural brain changes.

**Keywords:** Brain MRI, Generative Modeling, Longitudinal Modeling, Autoencoders, Brain Aging

## 1. Introduction

We consider the problem of generative modeling of a series of structural brain MRIs to predict how a subject's brain ages structurally. Faithful prediction of brain aging trajectories based on neuroimaging modalities have been used to detect anomalous brain atrophy patterns (Chouliaras and O'Brien, 2023).

There is a growing body of work using deep generative models for 3D brain generation (Kwon et al.; Pombo et al., 2021; Pinaya et al., 2022), and particularly, structural aging (Ravi et al., 2022; Zhao et al.; Yoon et al., 2023; Jung et al., 2023). However, current works for 3D brain aging either involve complicated training routines (Ravi et al., 2022) and hyperparameter tuning or are hard to scale up to high resolution (Yoon et al., 2023), unless slice-wise generation is used (Ravi et al., 2022).

In this work, we propose MRExtrap, a simple and effective method for simulating 3D brain aging trajectories on the voxel level. We discover that training a convolutional autoencoder to compress T1 MRI volumes $x \in \mathbb{R}^{D \times D \times D}$ yields a latent space $z \in \mathbb{R}^{4 \times d \times d \times d}$ that has allows linear predictions: In particular, in latent space, we find linear relationship between key regional brain volumes $v = \text{Segment}(x)$ and latent codes $z$ within a subject (see Fig. 1a). Using sequences of T1 MRIs from the ADNI database, we perform linear regression to predict $z$ from age $a \in \mathbb{R}^+$, and find that this simple approach allows us to predict brain regions from the extrapolated latents with high accuracy.

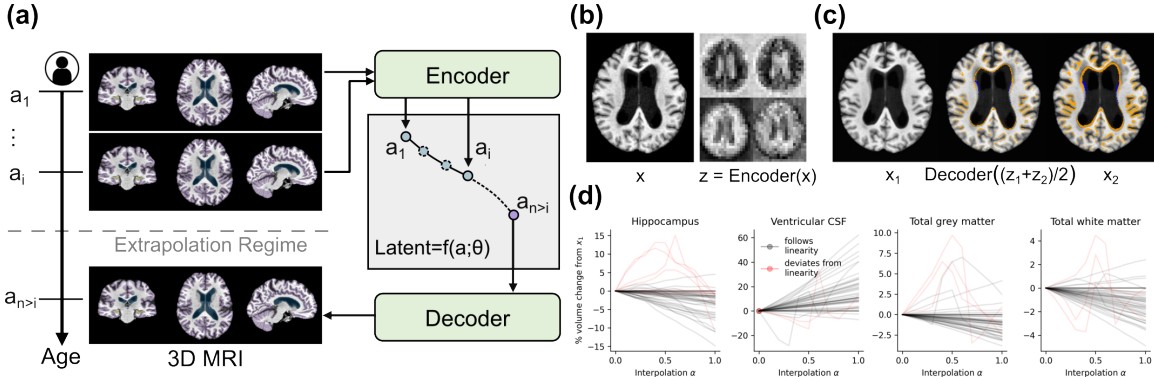

Figure 1: **MRExtrap for longitudinal brain MRI aging. (a)** We encode individual brain MRI volumes into a compressed latent space using a convolutional autoencoder, trained with the loss in Eq. (1). We predict multidimensional latents using age using linear regression and then decode the latents back into voxel space for an unseen age. **(b)** The encoded MRIs are spatially similar to the original volumes. **(c)** We find that a linear interpolation of two latents of a subject, when decoded, results in smoothly varying regional volumes. **(d)** Across subjects, volume changes in key brain regions are linear with respect to the interpolation factor $\alpha$, although not all subjects exhibit this behaviour (in red).

## 2. Methods and Results

We train an autoencoder to compress 3D MRIs $x \in \mathcal{X}$ to a latent representation $z \in \mathcal{Z}$ using the same methodology as in Rombach et al. (2022) and Pinaya et al. (2022). We use a combination of losses to train this autoencoder to reconstruct $x$, including $L_1$ reconstruction loss, a perceptual loss proposed in Zhang et al. (2018), and a patch-based adversarial discriminator (Isola et al., 2016). Further, we perform regularisation in latent space, obtaining the total loss

$$L_{\mathrm{AE}} = \|x - \hat{x}\|_1 + \beta_1 \mathrm{LPIPS}(x, \hat{x}) + \beta_2 \mathrm{AdvLoss}(x, \hat{x}) + \beta_3 D_{KL}(\mathrm{Enc}_{\mu,\Sigma}(x), \mathcal{N}(0, I)), \quad (1)$$

where $\beta_1, \beta_2, \beta_3$ are loss scaling terms, $\hat{x} = \mathrm{Dec}(\mathrm{Enc}_\mu(x))$, and $\mathrm{Enc}_{\mu,\Sigma}$ parameterizes the normal distribution of $z$. Since this training only requires individual volumes, we can use cross-sectional data as well as longitudinal data.

The latent space of the trained autoencoder exhibits useful interpolation properties. Specifically, we found that, within a subject, linearly interpolating between two latents $z_1$ and $z_2$ corresponding to 3D images $x_1$ and $x_2$ and with segmentation volumes $v_1, v_2 \in \mathbb{R}^+$, leads to $x_{\mathrm{interp}}$ with a segmentation volume $v_{\mathrm{interp}}$ that is a linear combination of $v_1, v_2$:

$$v_{\mathrm{interp}} = \mathrm{Seg}(\hat{x}_{\mathrm{interp}}) \approx \alpha v_1 + (1 - \alpha)v_2, \text{ where}$$
$$\hat{x}_{\mathrm{interp}} = \mathrm{Dec}(\alpha \mathrm{Enc}_\mu(x_1) + (1 - \alpha)\mathrm{Enc}_\mu(x_2)). \quad (2)$$

Here, the scalar volumes $v$ are obtained by segmenting the MRI and computing the voxels contained in each segmented region. In addition to exhibiting linearity, this interpolation preserves the subject-specific morphological structure (Fig. 1c,d) for the hippocampus, ventricle, GM, and WM volumes. This suggests that the relationship between age and

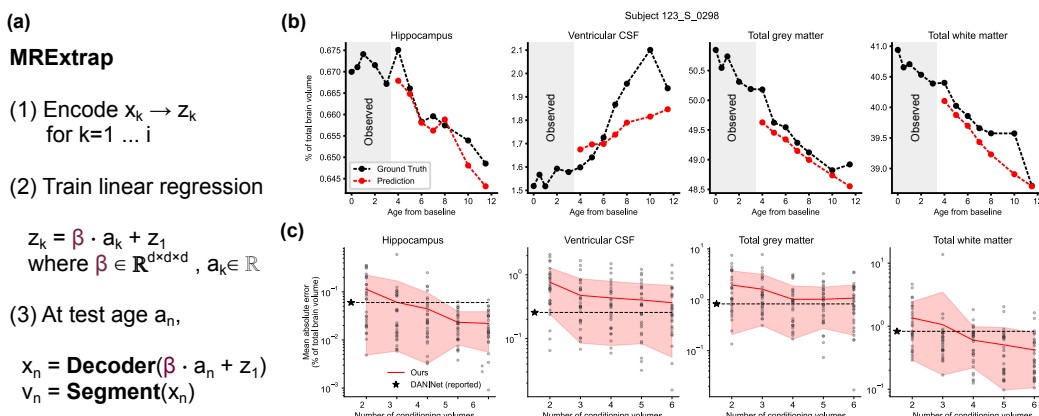

Figure 2: **Brain aging performance of MRExtrap on subjects in the ADNI dataset.** **(a)** MRExtrap admits a trained autoencoder and a sequence $x_1 \ldots x_i$ of longitudinal volumes per subject and predicts 3D MRI at test age $a_n > a_i$. **(b)** For an example subject 123_S_0298, linear regression from age to latents leads to accurate forecasting of regional brain volumes in voxel space. **(c)** Model accuracy improves when we consider more volumes for the regression. Linear regression using multiple volumes is competitive with Ravi et al. (2022).

structural brain volumes may be modelled using a linear relationship in the latent space (Salat et al., 2004). Thus, in order to predict a subject's future brain changes as a function of age, we fit a linear regression to the latents of that subject (algorithm in Fig. 2a).

To evaluate our method, we preprocessed 9200 T1 MRIs from 1700 patients from the ADNI database by applying affine registration (3+3 d.o.f) to the MNI atlas, followed by skull stripping. We then trained the autoencoder on 1500 patients for 230 epochs with $\beta_1 = 1, \beta_2 = 10^-3, \beta_3 = 10^{-6}$. We only applied the adversarial loss after 100 epochs to avoid training instability. Once trained, the autoencoder was frozen, and we used the Synthseg network (Billot et al., 2023) for segmentating the hippocampus, ventricular cerebrospinal fluid (CSF), as well as total gray and white matter (GM & WM) for each image. We found that on the test subjects, MRExtrap accurately forecasts the changes in the regional volumes for hippocampus, Ventricular CSF, and total GM and WM when conditioned on the first few volumes (Fig. 2b). Furthermore, the simpler linear regression prediction is competitive with a GAN-based baseline DANINet by Ravi et al. (2022). In contrast to DANINet, MRExtrap can use a variable number of MRIs as input – the forecasting error decreases as we consider more volumes for the regression (Fig. 2c).

In this work, we performed linear regression in the autoencoder latent space and showed its effectiveness in modeling structural aging patters for high-dimensional 3D brain MRIs. Our work indicates that modeling brain aging trajectories in latent spaces is an interesting and potentially fruitful research direction. Currently, our model requires at least two longitudinal scans for brain aging. As follow-up to this work, we will explore a combination of our linear regression approach with a neural network-based prior that will allow predictions conditioned on only one baseline scan.

**Acknowledgements** This work was funded by DFG EXC 2064/1, Project no. 390727645, and BMBF: Tübingen AI Center, FKZ: 01IS18039A. We thank IMPRS-IS for their support.

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
