# OpenReview forum: "MRExtrap: Linear Prediction of Brain Aging in Autoencoder Latent Space of MRI Scans"
_MIDL.io/2024/Short_Papers — MIDL 2024 Short Papers_

### Official Review · Reviewer_7JEX · 2024-04-24

**Confidence:** 3
**Final Rating:** 5

**Review:**

The authors use a VAE with linear projection in the latent space to predict images under brain aging behaviour. This is pretty neat, as one would expect that if the images themselves are modelled by the VAE, the latent space trajectory is not linear -- linearity makes it easy to project forward (And even backward?) . The authors compare to other image generation behaviour, and find comepetitive behaviour with Ravi et al, where a more sophisticated model is used.

The obvious comparison, which to me is missing (although it's just a short paper!) is to look at how such an embedding *of deformations* would work -- since a linear projection of deformations might capture the same thing, but be more interpretable and enable anatomical spatial analysis. Before the deep learning era (say 2018+), there were some linear models that that did predictive modeling of anatomy by doing linear predictions in deformation projections. Although surely they were not as powerful as using deep nets and VAEs, they showed promise.

This paper should lead to interesting discussion!

---

### Decision · Program_Chairs · 2024-04-26

Accept